# Inhibitory Potential of *Bifidobacterium longum* FB1-1 Cell-Free Supernatant against Carbapenem-Resistant *Klebsiella pneumoniae* Drug Resistance Spread

**DOI:** 10.3390/microorganisms12061203

**Published:** 2024-06-14

**Authors:** Jing Wang, Dan-Cai Fan, Rui-Shan Wang, Yu Chang, Xue-Meng Ji, Xin-Yang Li, Yan Zhang, Jing-Min Liu, Shuo Wang, Jin Wang

**Affiliations:** Tianjin Key Laboratory of Food Science and Health, School of Medicine, Nankai University, Tianjin 300071, China; wangjing77202203@163.com (J.W.); fandancai@nankai.edu.cn (D.-C.F.); w18103658595@163.com (R.-S.W.); 16630831938@163.com (Y.C.); jixuemeng@nankai.edu.cn (X.-M.J.); lixinyang@nankai.edu.cn (X.-Y.L.); yzhang@nankai.edu.cn (Y.Z.); liujingmin@nankai.edu.cn (J.-M.L.)

**Keywords:** carbapenem-resistant *Klebsiella pneumoniae*, *B. longum*, antimicrobial activity, resistance genes, plasmid conjugation transfer

## Abstract

The widespread dissemination of carbapenem-resistant *Klebsiella pneumoniae* (CRKP) and its drug resistance transfer poses a global public health threat. While previous studies outlined CRKP’s drug resistance mechanism, there is limited research on strategies inhibiting CRKP drug resistance spread. This study investigates the potential of *Bifidobacterium longum* (*B. longum*) FB1-1, a probiotic, in curbing the spread of drug resistance among CRKP by evaluating its cell-free supernatant (CFS) for antibacterial activity. Evaluating the inhibitory effect of FB1-1 CFS on CRKP drug resistance spread involved analyzing its impact on drug resistance and virulence gene expression; drug resistance plasmid transfer FB1-1 CFS exhibited an MIC range of 125 μL/mL against CRKP. After eight hours of co-culture, CFS achieved a 96% and 100% sterilization rate at two and four times the MIC, respectively. At sub-inhibitory concentrations (1/2× MIC), FB1-1 CFS reduced the expression of the *bla_KPC* gene, which is pivotal for carbapenem resistance, by up to 62.13% across different CRKP strains. Additionally, it markedly suppressed the expression of the *uge* gene, a key virulence factor, by up to 91%, and the *fim_H* gene, essential for bacterial adhesion, by up to 53.4%. Our study primarily focuses on determining the inhibitory effect of FB1-1 CFS on CRKP strains harboring the *bla_KPC* gene, which is a critical resistance determinant in CRKP. Furthermore, FB1-1 CFS demonstrated the ability to inhibit the transfer of drug resistance plasmids among CRKP strains, thus limiting the horizontal spread of resistance genes. This study highlights FB1-1 CFS's inhibitory effect on CRKP drug resistance spread, particularly in strains carrying the *bla_KPC* gene, thus offering a novel idea and theoretical foundation for developing antibacterial drugs targeting CRKP resistance.

## 1. Introduction

*Klebsiella pneumoniae*, an acquired hospital infection pathogen, is linked to severe conditions like pneumonia, wound infections, urinary tract infections, and septicemia [1,2]. Primary drugs for treating *Klebsiella pneumoniae* are β-lactam antibiotics [3], though prolonged antibiotic exposure leads to gene mutations and β-lactamase secretion, rendering them ineffective through hydrolysis [4,5]. Widespread carbapenem-resistant *Klebsiella pneumoniae* dissemination is mainly attributed to the horizontal transfer of mobile genetic elements, especially transposons encoded in insertion sequences and plasmids [6,7]. The *bla_KPC* gene, commonly carried on transferable plasmids within the conserved Tn3 family transposon Tn4401, facilitates persistent and effective drug resistance spread through horizontal transfer [8]. Advanced drug-resistant pathogen treatment methods include phage therapy, gene editing, and antibody therapy [8,9,10,11]. Phage therapy safety has long been controversial due to phages’ inherent toxicity [12]. Additionally, the high costs and limited success rates of gene editing, and the short shelf life and production costs of antibody therapy hinder their practical application [13,14]. Consequently, identifying new safe alternative therapies for drug-resistant pathogens is urgently needed.

Probiotics, active microorganisms, have beneficial effects on the human body when administered adequately [15]. Research has demonstrated that probiotics can prevent gastrointestinal infections through probiotics’ production of antibacterial substances that stimulate host immunity [16,17,18]. Furthermore, the functional metabolites, including organic acids and antimicrobial peptides produced by probiotics, effectively compete with pathogens [19,20,21], preventing diseases. Liu et al. [22] showed that *Lactobacillus rhamnosus* SHA113 cultures, both living cells and supernatants, effectively inhibit multidrug-resistant *Staphylococcus aureus*, suggesting their potential as antibacterial agents. Scillato et al. [23] demonstrate significant inhibitory effects of living cells and cell-free supernatant of *Lactobacillus gasseri* 1A-TV, *L. fermentatum* 18a-TV, and *L. crispatus* 35a-TV against multidrug-resistant urogenital pathogens, such as *Escherichia coli*, *Staphylococcus aureus*, *Candida albicans*, and *Enterococcus*. These findings highlight the potential of these probiotic formulations against multidrug-resistant pathogens. However, limited research exists on the antibacterial properties and mechanism of *Bifidobacterium*.

In this study, we isolated multiple strains of *B. longum* from the fecal samples of healthy breast-fed infants. Through bacteriostatic experiments, we identified a potent strain, *B. longum* FB1-1, which exhibited strong bacteriostatic activity. We conducted an in-depth investigation into the mechanism by which the CFS of *B. longum* FB1-1 inhibits drug resistance in *Klebsiella pneumoniae*. Furthermore, we performed qualitative and quantitative metabolic analyses to elucidate the antibacterial substances produced by *B. longum* FB1-1.

## 2. Materials and Methods

### 2.1. Isolation and Genomic Analysis

The collected infant feces samples were diluted with sterile normal saline to create a concentration gradient. The diluted samples were then applied to a solid screening me-dium and incubated for 24–48 h at 37 °C in a conventional incubator and strictly anaerobic conditions until colony growth occurred. The colonies suspected of being *Bifidobacterium* were chosen, and subjected to multiple purification steps on a screening culture medium, and the strain retaining the characteristics of the suspected *Bifidobacterium* colonies was selected as the target strain after repeated purification. The preparation of the solid screening culture for *Bifidobacterium* involved augmenting the MRS culture medium (Becton, Dickinson and Company, Franklin Lakes, NJ, USA) with 50 mL of filtered equine serum (Thermo Fisher Scientific, Waltham, MA, USA), 0.05% L-cysteine hydrochloride (Sigma-Aldrich, St. Louis, MO, USA), and 0.05 mg/mL of mupirocin (Sigma-Aldrich, St. Louis, MO, USA) per liter. L-cysteine hydrochloride underwent sterile filtration for aseptic treatment, and mupirocin was added post autoclaving, once the culture medium had cooled to 40–50 °C.

### 2.2. Antibacterial Activity

#### 2.2.1. Determination of Antibacterial Activity

Carbapenem-resistant *Klebsiella pneumoniae* ATCC BAA-1705, BNCC358281, and BNCC289979 (Beina Biology Co., Ltd., Beijing, China) underwent 12–24 h of incubation at 37 °C and 200 rpm in Luria-Bertani (LB) medium (Becton, Dickinson and Company, Franklin Lakes, NJ, USA) for the reactivation of CRKP.

To preliminarily evaluate the antibacterial activity of FB1-1 CFS against CRKP, the CRKP culture was first grown to the logarithmic phase, and its optical density at 600 nm (OD600) was adjusted to 0.1–0.2. The standardized CRKP culture was then evenly spread on LB agar plates. An Oxford cup (a small, cylindrical container made of stainless steel or plastic, commonly used in microbiology laboratories to test the sensitivity of bacteria to various antibiotics; Sigma-Aldrich, St. Louis, MO, USA) was employed for the assay. An aliquot of the complete bacterial culture of *B. longum*, grown to the logarithmic phase, was dispensed into each well of the Oxford cup, while MRS liquid culture medium served as a control. Following an 18–24 h incubation at 37 °C in a constant-temperature incubator (Thermo Fisher Scientific, Waltham, MA, USA), the diameters of the zones of inhibition were measured. The assay was performed in triplicate to ensure reproducibility.

#### 2.2.2. Determination of Minimum Inhibitory Concentration (MIC)

The MIC of FB1-1 CFS against various CRKP strains was determined using the broth microdilution method according to CLSI (Clinical and Laboratory Standards Institute, Wayne, PA, USA) standards. CRKP strains were grown overnight on blood agar plates (BD Biosciences, Franklin Lakes, NJ, USA) at 37 °C, and the bacterial suspension was adjusted to 0.5 McFarland standard using sterile saline (Baxter Healthcare Corporation, Deerfield, IL, USA). The suspension was then diluted to 1:100 in LB broth to achieve a final concentration of approximately 5 × 10^5^ CFU/mL. Two-fold serial dilutions of the CFS were prepared in LB, and 100 µL of each dilution was added to the wells of a 96-well microtiter plate (Thermo Fisher Scientific, Waltham, MA, USA). An equal volume of the prepared bacterial inoculum was added to each well, resulting in a final volume of 200 µL per well and a bacterial concentration of approximately 5 × 10^5^ CFU/mL. Positive (bacterial inoculum without CFS) and negative (LB broth only) control wells were included. The plate was covered and incubated at 37 °C for 18–24 h. The MIC was determined as the lowest concentration of CFS that completely inhibited visible bacterial growth. The experiment was conducted in triplicate.

#### 2.2.3. Bacteriostasis and Sterilization

To assess CRKP growth in the presence of FB1-1 CFS, bacteriostatic and bactericidal effects were examined. CRKP overnight cultures were adjusted to OD600 = 0.2 in fresh LB medium. FB1-1 CFS was subsequently added to bacterial suspension at final concentrations of 0, 1/4× MIC, 1/2× MIC, 1× MIC, and 2× MIC, with a total volume adjusted to 200 μL in a 96-well plate. CRKP growth was measured at the OD600 after 2, 4, 6, 8, 10, and 12 h of incubation. The experiment was conducted in triplicate.

To determine the bactericidal effect of FB1-1 CFS on CRKP, a time-kill assay was performed. A standardized bacterial suspension of CRKP was prepared and adjusted to a concentration of 1 × 10^6^ CFU/mL in LB broth. FB1-1 CFS was introduced to the bacterial suspension for co-culturing at final concentrations of 0 (control), MIC, 2× MIC, and 4× MIC. The co-cultures were incubated at 37 °C with shaking at 200 rpm. At time points of 2, 4, and 6 h, aliquots of the co-cultures were collected, serially diluted, and plated on solid LB agar plates. Following overnight incubation at 37 °C, viable bacteria were quantified by counting the number of colonies on the plates. The colony counts were used to calculate the log10 CFU/mL at each time point. The experiment was conducted in triplicate, and the mean log10 CFU/mL values were plotted against time to generate the time-kill curves. The bactericidal effect of FB1-1 CFS was evaluated by comparing the reduction in bacterial population at different concentrations over time, with a ≥3 log10 CFU/mL reduction considered as bactericidal activity.

#### 2.2.4. Inhibition of Gene Expression

The inhibitory effects of FB1-1 CFS on the expression of CRKP resistance gene *bla_KPC*, virulence-associated gene *uge*, and bacterial adhesion-related gene *fim_H* were assessed. First, LB broth medium was used to adjust the OD600 of CRKP to 0.1–0.2 during the logarithmic phase FB1-1 CFS, adjusted to MIC, incubated with CRKP for 18–24 h at 37 °C, and served as the experimental condition, while the control lacked CFS. Subsequently, the total RNA was extracted from both the co-culture and control groups using a total RNA extraction kit (Qiagen, Hilden, Germany) and subjected to reverse transcription. Reverse transcription quantitative real-time polymerase chain reaction (qRT-PCR) assessed the relative expressions of CRKP drug resistance and virulence genes (as presented in Table 1). The qRT-PCR was performed using a QuantStudio 5 Real-Time PCR System (Thermo Fisher Scientific, Waltham, MA, USA) and SYBR Green PCR Master Mix (Applied Biosystems, Foster City, CA, USA). The experiment was conducted in triplicate.

#### 2.2.5. Plasmid Junction Transfer

In order to elucidate the influence of FB1-1 CFS on the intercellular transfer of plasmids conferring resistance in CRKP, we employed the methodological framework outlined by Weng et al. [24]. To activate the donor and recipient strains for conjugation experiments, CRKP, resistant to meropenem (Sigma-Aldrich, St. Louis, MO, USA) and rifampin (Sigma-Aldrich, St. Louis, MO, USA), was selected as the donor, while *E. coli* EC600 (Bao Sai Plasmid Co., Ltd., Hangzhou, China), resistant to rifampin and meropenem, served as the recipient. Single colonies of the donor and recipient strains were inoculated into LB broth and incubated at 37 °C with shaking at 120 rpm for 14 hours. LB agar plates containing meropenem (1 μg/mL) and rifampin (1 μg/mL) were prepared for selective growth of the donor strain, while plates containing meropenem (1 μg/mL) and rifampin (1 μg/mL) were used for the recipient strain, with antibiotic concentrations determined based on the MICs of the respective strains. After incubation, the cultures were diluted to 1:100 in fresh LB broth and further incubated until an optical density at OD600 of 0.6 was reached, corresponding to the mid-logarithmic growth phase. The activated donor and recipient cultures were then used for subsequent conjugation experiments, with specific antibiotic concentrations and growth conditions maintained to ensure selective pressure and facilitate the transfer of drug resistance genes between the strains.

Subsequently, the CFS of FB1-1 was serially diluted in LB broth to achieve final concentrations of 0 μL/mL, 31.25 μL/mL, 62.50 μL/mL, and 125 μL/mL. It was determined that the MIC required to impede the growth of CRKP was 125 μL/mL. Consequently, dilutions to concentrations of 31.25 μL/mL and 62.50 μL/mL did not exert an inhibitory effect on the growth of CRKP. The activated donor and recipient bacteria were then co-inoculated at a 4:1 ratio into LB medium containing different FB1-1 CFS concentrations and incubated for 4 h at 37 °C. Following the culture, the transconjugant was appropriately diluted and applied with a coating rod onto LB solid agar plates containing meropenem and rifampicin to screen for the transconjugant. After full absorption of the bacterial liquid, the plates were inverted and cultured overnight at 37 °C for 18–24 h, followed by the counting of single colonies. The experiment was conducted in triplicate.

The bacterial solution underwent plasmid DNA extraction using a small plasmid extraction kit from Tiangen Biotechnology Co., Ltd. (Beijing, China). Amplification of plasmid *bla_KPC* DNA utilized a PCR kit from Ningbo Baichuan Biotechnology Co., Ltd., with synthesized primers (5′-CGTCTAGTTCTGCTGTCTTG-3′ [forward] and 5′-CTTGTCATCCTTGTTAGGCG-3′ [reverse]) from Beijing Qingke Biology Co., Ltd., Beijing, China. The PCR program included initial denaturation at 94 °C for 2 min, followed by denaturation at 94 °C for 30 s, annealing at 55 °C for 30 s, extension at 72 °C for 30 s, and a final extension at 72 °C for 2 min. Detection of amplified *bla_KPC* DNA was achieved through 1% agarose gel electrophoresis (Biowest, Nuaillé, France). The gel was visualized using a Gel Doc XR+ System (Bio-Rad Laboratories, Hercules, CA, USA) and the Image Lab Software (version 6.0.1, Bio-Rad Laboratories, Hercules, CA, USA).

### 2.3. Gene Sequencing and Annotation of Bifidobacterium

The *B. longum* FB1-1 genomic DNA was collected, purified, and fragmented using Covaris to construct a genomic sequencing library. The genomic DNA concentration was analyzed using TBS380 or Nanodrop2500 to ensure appropriate DNA quality for subsequent experiments (no degradation, OD260/280 = 1.8–2.0, total not less than 10 μg). The Illumina second-generation sequencing platform was employed to construct fragments with an insertion length of −400 bp from quality-checked DNA samples. PE150 (pair-end) sequencing, with a single-end sequencing reading length of 150 bp, was executed. For bacterial genome scanning, short sequence assembly software Soap de Novo 2 (version 2.04, http://soap.genomics.org.cn/, accessed on 10 July 2023) spliced multiple Kmer parameters of the optimized sequence post-second-generation sequencing to obtain optimal contig assembly results. This was followed by comparing reads to contig, conducting local assembly, and optimizing the assembly result based on paired-end and overlap relationships of read to form scaffolds. The bacterial genome completion map was prepared via three-generation sequence assembly using the assembly software unicycler R (version 0.4.8, https://github.com/rrwick/Unicycler/releases, accessed on 10 July 2023). During the assembly process, sequence correction was performed using pilon software (version 1.24, https://github.com/broadinstitute/pilon, accessed on 10 July 2023). If there was overlap with a certain length or more at both ends of the final assembly sequence, the sequence was looped, and the overlap sequence at one end truncated. Finally, a complete chromosome and plasmid sequence were obtained.

Using Glimmer (version 3.02, http://ccb.jhu.edu/software/glimmer/index.shtml, accessed on 10 July 2023), GeneMarkS (version 4.3, http://topaz.gatech.edu/GeneMark, accessed on 10 July 2023) and Prodigal software (version 2.6.3, https://github.com/hyattpd/Prodigal, accessed on 10 July 2023), the coding sequences (CDSs) in the genome were predicted. Prodigal was employed by default for the assembly results of scanned images, while, for completed images, Prodigal was utilized for chromosomal genomes, and GeneMarkS was applied for plasmid genomes. To predict tRNA within the genome, TRNA scan-SEv2.0 software (version 2.0.12, http://trna.ucsc.edu/software/, accessed on 10 July 2023) was employed, providing nucleotide sequence information, anticodon details, and secondary structure information for each sample genome’s tRNA. Barrnap software (version 0.9, https://github.com/tseemann/barrnap, accessed on 10 July 2023) was used to predict the rRNA in the genome, yielding information on the type, location, and sequence of all rRNA in each sample genome. Comparisons between the predicted coding genes and the Clusters of Orthopedic Groups from the Kyoto Encyclopedia of Genes were performed in order to annotate the protein functions.

### 2.4. Analysis of Antibacterial Compounds

The micromolecular compounds in the CFS of *B. longum* FB1-1 were identified and qualitatively analyzed by LC-MS (liquid chromatography-mass spectrometry) to ascertain the antibacterial substance components. Initially, a 100 mg mixed sample was placed in a 2 mL centrifuge tube. Next, 1 mL of 70% methanol and 3 mm steel balls were added, and the mixture was shaken for 3 min using a fully automated sample rapid grinder (JXFSTPRP-48, 70 Hz; Shanghai Jingxin Industrial Development Co., Ltd., Shanghai, China). Subsequently, the sample underwent cooling and sonication (40 kHz) for 10 min. The resulting supernatant was centrifuged at 12,000 rpm at 4 °C for 10 min and diluted 2-100 times. Finally, 10 μL of a 100 μg/mL internal standard was added and tested using a 0.22 μm PTFE filter.

The supernatant was initially prepared at concentrations of 4× MIC, 2× MIC, MIC, and 1/2× MIC, and subsequently subdivided into four equal volumes for distinct experimental interventions. The control cohort consisted of the supernatant in its untreated form, serving as a baseline for comparative analysis. Experimental Group A was subjected to a thermal inactivation process, where the samples were autoclaved at 121 °C for a duration of 15 min. Experimental Group B underwent a pH adjustment procedure, wherein NaOH (Sigma-Aldrich, St. Louis, MO, USA) was utilized to achieve a neutral pH, exploring the effect of pH modulation on antibacterial activity. For Experimental Group C, a more nuanced approach was employed to investigate the role of proteinaceous substances in the antibacterial activity observed. Specifically, this group’s supernatant was treated with protease K at a concentration of 2 mg/mL and incubated at 37 °C for 2 h to facilitate enzymatic digestion. Subsequently, to inactivate the protease K (Sigma-Aldrich, St. Louis, MO, USA), the samples were subjected to a brief heat treatment at 100 °C for 2 min. Following a 12-hour co-culturing period with CRKP, the antibacterial activity of the control and experimental groups was quantitatively assessed by measuring the OD600. The experiment was conducted in triplicate.

To further substantiate whether the antimicrobial action of FB1-1 CFS CRKP is attributable to pH alterations, a meticulous experimental design was implemented. Initially, the pH of FB1-1 CFS was precisely measured, establishing a reference point for the ensuing experimental manipulations. Two sets of MRS broth were prepared: one was pH-adjusted to align with that of FB1-1 CFS utilizing phosphate buffer systems, while the other remained unmodified, serving as a baseline for comparison. The overnight-cultured CRKP bacterial suspension was adjusted to an OD600 of 0.1–0.2. CRKP isolates were then co-cultured in both the pH-adjusted and unmodified MRS broths, to discern the bacteriostatic effects. CRKP bacterial suspension was also co-cultured with untreated FB1-1 CFS as a positive control, and separately cultured with an equivalent proportion of LB broth as a negative control. CRKP growth was measured at OD600 after 2, 4, 6, and 8 h of incubation. The experiment was conducted in triplicate.

To further ascertain whether the antimicrobial efficacy of FB1-1 CFS is solely attributed to its highest concentration of two antimicrobial compounds, citric acid and epirizole (both from Sigma-Aldrich, St. Louis, MO, USA) were diluted to their MIC (125 μL/mL) and their antibacterial activity was assessed in comparison with FB1-1 CFS using the Oxford cup method. The experiment was conducted in triplicate.

### 2.5. Statistical Analysis

All statistical analyses were performed using GraphPad Prism software version 9.4.1 (GraphPad Software, San Diego, CA, USA, https://www.graphpad.com/, accessed on 16 December 2023). Data were first tested for normality using the Shapiro–Wilk test. For normally distributed data, comparisons between two groups were conducted using an unpaired two-tailed Student’s *t*-test, whereas one-way ANOVA followed by Tukey’s multiple comparisons test was applied for multiple group comparisons. The *p*-value for statistical significance for all analysis is defined as *p* < 0.05.

## 3. Results

### 3.1. Antibacterial Activity

Three strains of *B. longum* were isolated from infant feces and designated as FB1-1, FB1-2, and FB1-3. The antibacterial activity of their supernatant was assessed using the Oxford cup method. Figure 1A illustrates the results obtained, comparing the control group with the experimental group treated with FB1-1 CFS. It is evident that the experimental group exhibited a clear inhibition zone. Among the five bacteriostatic circles displayed, *B. longum* FB1-1 CFS demonstrated the most pronounced bacteriostatic effect. The diameters of the inhibition zones of BAA-1705, BNCC358281, and BNCC289979 were 21.4 cm, 19.8 cm, and 19.9 cm, respectively (Oxford cup experimental phenomena of BAA-1705, BNCC358281, and BNCC289979 are shown in the Appendix A). This finding suggests that *B. longum* FB1-1 CFS possesses significant antibacterial activity against CRKP. Consequently, FB1-1 CFS was selected for further investigation into its antibacterial mechanism. Figure 1B reveals that at, a concentration of 62.5 μL/mL, the CFS exhibited no inhibitory effect on the growth of CRKP. Therefore, the MIC of the CFS of *B. longum* FB1-1 on CRKP was determined to be 125 μL/mL. (The MIC for BAA-1705, BNCC358281, and BNCC289979 was also determined to be 125 μL/mL in triplicate.) The growth curve of CRKP in response to various CFS concentrations is presented in Figure 1C. At concentrations of 1/8×, 1/4×, and 1/2× MIC, the CFS from *B. longum* FB1-1 minimally impacted CRKP growth. Conversely, at the MIC concentration, complete inhibition occurred within the initial 10 h of co-culturing. Notably, when the CFS concentration was twice the MIC, CRKP growth was entirely suppressed. The sterilization rate of FB1-1 CFS at different concentrations was determined (Figure 1D). At the MIC, the average sterilization rate of CRKP was 90.27%. Significantly, doubling the CFS concentration improved the average sterilization rate to 96% and, at four times the MIC, CRKP growth was completely inhibited, resulting in a sterilization rate of 100%. These findings unequivocally demonstrate the substantial inhibitory effect of FB1-1 CFS on CRKP growth.

Figure 1B reveals that at a concentration of 62.5 μL/mL, the CFS exhibited no inhibitory effect on the growth of CRKP. Therefore, the MIC of the CFS of *B. longum* FB1-1 on CRKP was determined to be 125 μL/mL. (The MIC for BAA-1705, BNCC358281, and BNCC289979 was also determined to be 125 μL/mL in triplicate.) The growth curve of CRKP in response to various CFS concentrations is presented in Figure 1C. At concentrations of 1/8×, 1/4×, and 1/2× MIC, the CFS from *B. longum* FB1-1 minimally impacted CRKP growth. Conversely, at the MIC concentration, complete inhibition occurred within the initial 10 h of co-culturing. Notably, when the CFS concentration was twice the MIC, CRKP growth was entirely suppressed. The sterilization rate of FB1-1 CFS at different concentrations was determined (Figure 1D). At the MIC, the average sterilization rate of CRKP was 90%. Significantly, doubling the CFS concentration improved the average sterilization rate to 96% and, at four times the MIC, CRKP growth was completely inhibited, resulting in a sterilization rate of 100%. These findings unequivocally demonstrate the substantial inhibitory effect of FB1-1 CFS on CRKP growth.

### 3.2. Expression of Drug Resistance and Virulence Genes

This study’s findings, as depicted in Figure 2, extend to three strains of CRKP treated with FB1-1 CFS. The BAA-1705 strain exhibited a notable 32% reduction in *bla_KPC* gene expression and an extraordinary 91% suppression of the *uge* gene, which is a significant determinant of virulence. The *fim_H* gene, crucial for bacterial adhesion, showed a substantial 53.4% decrease. The BNCC358281 strain demonstrated an even more pronounced 62.13% reduction in *bla_KPC* gene expression, with a 15.44% decrease in *fim_H* and a 12.29% suppression of the *uge* gene. The BNCC289979 strain also showed significant suppression, with a 52.28% reduction in *bla_KPC*, a more substantial 31.49% decrease in *fim_H*, and a 17.71% suppression of *uge*. These results collectively highlight the inhibitory effect of FB1-1 CFS on the expression of drug resistance and virulence genes across the strains of CRKP studied, suggesting a potential therapeutic strategy against CRKP infections.

### 3.3. Plasmid Binding Transfer

Figure 3A depicts the impact of varied FB1-1 CFS concentrations on plasmid transconjugant growth. The results show a negative correlation between FB1-1 CFS concentration and plasmid transconjugant growth, indicating higher concentrations lead to reduced growth. In Figure 3B, plasmid transconjugant growth is quantified with varying supernatant concentrations. The findings reveal a significant difference in growth numbers between experimental and control groups. Specifically, FB1-1 CFS exhibits potent inhibitory effects on plasmid transfer, with inhibition rates of 96.14% and 93.90% at the MIC and sub-inhibitory concentration, respectively. These results highlight the robust antibacterial capabilities of FB1-1, positioning it as a valuable agent for impeding plasmid binding transfer. In Figure 3C, using the marker as a reference, lane 1 represents the DNA banding pattern of the negative control donor bacteria *E. coli* EC600, while lane 2 displays the DNA banding pattern of CRKP. The prominent band observed at the 750 bp position within lane 2 indicates the successful transfer of the *bla_KPC* plasmid, which harbors a quintessential resistance gene commonly found in CRKP. This result signifies the effective horizontal gene transfer of the *bla_KPC* resistance determinant, underscoring its role in the dissemination of antibiotic resistance among bacterial populations.

### 3.4. Gene Sequence and Information Annotation

The genomic distribution of *B. longum* FB1-1 isolated from infant feces is illustrated in the circular genomic map (Figure 4A). The complete genomic sequence of FB1-1, a 2.41 Mb ring chromosome, slightly smaller than the reference strain *B. longum*, displayed no plasmid and a 60.24% GC content. FB1-1 comprised 2005 coding genes, totaling 2,101,452 bp, with an average length of 1048.11 bp, constituting 87.21% of the entire genome. Additionally, FB1-1 featured 56 tRNAs facilitating amino acid transport and 8 rRNAs (four 16S rRNAs and four 23S rRNAs). In terms of functional analysis using the KEGG database (https://www.genome.jp/kegg/, accessed on 13 December 2023) (Figure 4B), 1078 genes in FB1-1 were assigned to various pathways, predominantly in carbohydrate metabolism (115) and amino acid metabolism (115). Notably, FB1-1 encompassed 168 genes contributing to secondary metabolite biosynthesis, suggesting its potential probiotic function through the synthesis of these secondary metabolites.

The corresponding genetic information has been uploaded to the National Center for Biotechnology Information (NCBI) database. The RefSeq assembly accession number for this genomic sequence is GCF_037997085.1. This sequence has been made publicly available for the scientific community to ensure transparency and reproducibility of our findings. The assembly provides a comprehensive representation of the genomic architecture and has been annotated to include genes, proteins, and other functional elements.

### 3.5. Analysis of Antibacterial Substances

LC-MS data revealed that organic acids constituted 74.77% of the primary antibacterial components in the CFS of *B. longum* FB1-1. Notably, among the antibacterial substances detected, citric acid, epirizole, DL-arginine, and betaine exhibited higher relative abundances (as presented in Table 2) [25,26,27,28,29,30,31,32,33].

Figure 5A depicts experimental phenomena, showing that, although the inhibitory impact of high-temperature-treated CFS on CRKP was lower compared to the control, it still exhibited significant antibacterial activity, underscoring its potential as an antibacterial agent. Additionally, CFS with a neutral pH lacked bacteriostatic ability, while adding protease K solution conferred a bacteriostatic ability akin to the control. These results suggest that FB1-1 CFS effectively inhibits CRKP growth through organic acid secretion. Furthermore, FB1-1 CFS exhibited notable high-temperature resistance.

The antibacterial properties of FB1-1 CFS and their potential correlation with pH changes were evaluated, with the results presented in Figure 5B. The antibacterial activity of MRS broth adjusted to the pH of FB1-1 CFS was nearly identical to that of FB1-1 CFS itself, whereas unadjusted MRS broth exhibited marginal antibacterial activity within the first 2 h of co-culture, which was completely lost after 4 h. Therefore, the antibacterial characteristics of FB1-1 CFS can be attributed to the pH changes induced by acidic substances produced by FB1-1 CFS.

As illustrated in Figure 5C, the antibacterial effect of the FB1-1 CFS compound was verified. Citric acid in its pure form demonstrated some bacteriostatic activity, whereas epirizole in its pure form exhibited negligible inhibitory effects. FB1-1 CFS continued to display significant inhibitory effectiveness, suggesting that the bacteriostatic effect of FB1-1 CFS is the result of a synergistic action of various antimicrobial compounds it contains.

## 4. Discussion

Bacterial tolerance to antibiotics gradually emerges as a consequence of prolonged selection and competitive pressures, ultimately culminating in the evolution of resistance. Effectively overcoming resistance necessitates targeting the virulence traits and resistance mechanisms exhibited by pathogens. The horizontal transfer of plasmids carrying resistance determinants serves as a predominant catalyst for the rapid and extensive dissemination of CRKP [8,9,10,11]. Therefore, the development of strategies aimed at impeding the horizontal transfer of resistance plasmids in CRKP holds the potential to fundamentally curb the proliferation of bacterial resistance.

The experimental findings revealed a significant inhibitory effect of the acellular supernatant derived from *B. longum* FB1-1 on CRKP, as evidenced by an MIC of 125 μL/mL, representing a dilution ratio of 12.5% of the original supernatant concentration. Remarkably, this MIC value was found to be lower than those documented for the majority of probiotic supernatants investigated.

Previous studies have identified that the transfer of plasmids carrying *bla_KPC* is a significant factor in CRKP resistance’s global spread [34]. To our knowledge, no reports currently exist on probiotic supernatants downregulating the *bla_KPC* gene in CRKP. Our research is inaugural to demonstrate that sub-inhibitory concentrations of FB1-1 CFS can markedly reduce the expression of key genes that contribute to the pathogenicity and antibiotic resistance of CRKP. In particular, the BAA-1705 strain showed a notable 32% reduction in *bla_KPC* gene expression, which is critical for β-lactam antibiotic resistance, and an extraordinary 91% suppression of the *uge* gene, a significant determinant of virulence. Additionally, the *fim_H* gene, which plays a crucial role in bacterial adhesion and colonization, was reduced by 53.4%. These results indicate that FB1-1 CFS not only targets resistance mechanisms but also significantly impairs the virulence and colonization capabilities of CRKP. Further analysis of the BNCC358281 and BNCC289979 strains revealed a more varied response, with significant reductions in *bla_KPC* gene expression by 62.13% and 52.28%, respectively. The variations in *fim_H* and *uge* gene suppression among these strains suggest a strain-specific response to FB1-1 CFS treatment, which could be due to genetic differences affecting the interaction between the bacterial cell and the bioactive compounds in the CFS.

Furthermore, Weng and colleagues previously demonstrated that baicalein, at 40 and 400 μg/mL, significantly inhibits CRKP plasmid transfer to *E. coli* EC600 [24]. Similarly, our study revealed that *B. longum* FB1-1 CFS significantly inhibits CRKP plasmid transfer to *E. coli* EC600 at concentrations of 31.25 and 125 μL/mL, marking the first report of probiotic supernatant inhibiting CRKP plasmid transfer.

In a previous study, probiotics demonstrated the ability to inhibit pathogenic bacterial growth through the release of acidic metabolites such as citric acid, lactic acid, and acetic acid. [35] Chen et al. [36] showed that organic acids in the supernatant of lactic acid bacteria significantly inhibited the activity of carbapenemase-resistant pathogenic bacteria. Lactic acid and mixed acids completely suppressed CRKP CPE0011 growth, while acetic acid exhibited a 67.8% inhibition rate against CPE0011. Wong et al. [37] identified oleic acid and myristic acid in the supernatant of *Lactobacillus rhamnosus* as key substances inhibiting drug resistance in *Staphylococcus aureus*. Our study highlights the intricate antibacterial mechanisms inherent in the FB1-1 CFS, emphasizing its dominant organic acid composition, which accounts for 74.77% of its primary antibacterial agents. Notably, the CFS of *B. longum* FB1-1 displays remarkable resistance to high temperatures, underscoring its potential utility in diverse antibacterial applications. Further experimentation showed that adjusting the pH of MRS broth to match that of FB1-1 CFS results in antibacterial activity almost identical to the native CFS. This finding demonstrates that the pH adjustment, driven by the acidic components within the CFS, is critical for its antibacterial efficacy, while unadjusted MRS broth only exhibits transient antibacterial effects. Our analysis also explored the impact of individual components within the CFS. Citric acid, for instance, contributes some bacteriostatic activity, whereas epirizole has minimal impact, suggesting a synergistic interplay among various compounds that enhances the overall antibacterial action.

It is crucial to acknowledge that the current study primarily focused on the inhibition of CRKP strains harboring the *bla_KPC* gene, which encodes for *Klebsiella pneumoniae carbapenemase* (*KPC*), a prevalent carbapenem-resistance determinant. To expand the applicability of these findings, future research should aim to investigate the inhibitory effects of FB1-1 CFS on CRKP strains carrying other clinically significant resistance genes, such as those encoding *metallo-β-lactamases* (*MBL*s) and *OXA-48-like carbapenemases*. These resistance mechanisms pose a severe threat to public health due to their ability to hydrolyze a broad spectrum of β-lactam antibiotics, including carbapenems, which are often considered last-resort antibiotics for treating multidrug-resistant infections.

## 5. Conclusions

In conclusion, this pioneering study has unveiled the remarkable potential of the CFS derived from *B. longum* FB1-1 as a novel therapeutic agent against CRKP strains harboring the *bla_KPC* gene. The significant inhibitory effects, low MIC, and the ability to downregulate key genes involved in pathogenicity and antibiotic resistance highlight the multifaceted mechanism of action of FB1-1 CFS. Furthermore, this study marks the first report of a probiotic supernatant inhibiting CRKP plasmid transfer, emphasizing its potential in curtailing the dissemination of resistance determinants. The elucidation of the intricate antibacterial mechanisms, particularly the predominance of organic acids and the synergistic interplay among various components, provides valuable insights for targeted interventions against drug-resistant pathogens. However, future research should investigate the inhibitory effects of FB1-1 CFS on CRKP strains carrying other clinically significant resistance genes, such as those encoding *MBL*s and *OXA-48-like carbapenemases*, to broaden the applicability of these findings. In summary, this groundbreaking study has unveiled the immense potential of *B. longum* FB1-1 CFS as a novel therapeutic agent against *bla_KPC*-positive CRKP strains, offering new perspectives on managing the escalating problem of antimicrobial resistance in clinical settings.

## Figures and Tables

**Figure 1 microorganisms-12-01203-f001:**
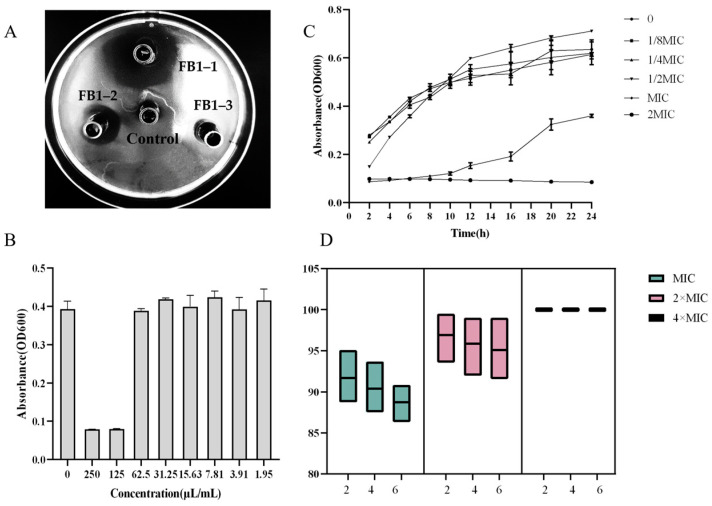
Inhibitory effect of *B. longum* CFS on CRKP(BAA-1705) growth. (**A**) Control group was cultured with MRS medium, while the experimental group was treated with *B. longum* CFS. (**B**) FB1-1 supernatant was diluted by 2 times gradient, while the growth of CRKP(BAA-1705) co-cultured with different concentration gradients of CFS was determined. The optical density (OD) of the CRKP suspension at 600 nm, which is a measure of bacterial concentration, was measured as the ordinate. (**C**) Determination of the growth of CRKP(BAA-1705) co-cultured with FB1-1 CFS of 0, 1/8× MIC, 1/4× MIC, 1/2× MIC, MIC, 2× MIC, and 4× MIC within 24 h. (**D**) Determination of sterilization rate of 2× MIC and 4× MIC FB1-1 CFS to CRKP(BAA-1705) within 6 h.

**Figure 2 microorganisms-12-01203-f002:**
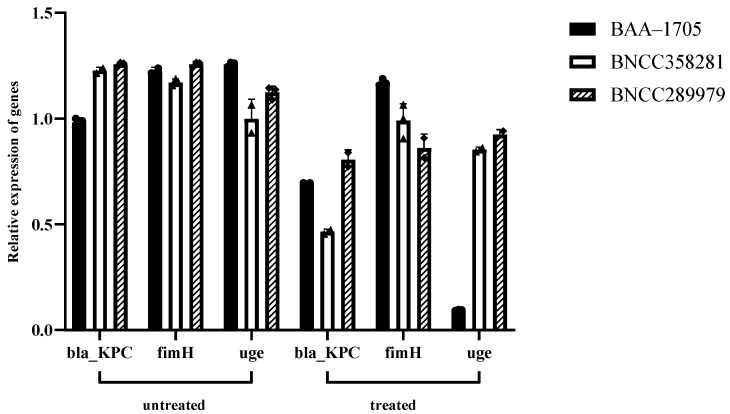
Inhibitory effect of CFS of FB1-1 on CRKP gene expression. Drug-resistant gene *bla_KPC*, virulence gene *fim_H*, *uge*. Treated group was co-cultured with CRKP at a sub-inhibitory concentration of 62.5 μL/mL, which did not affect CRKP growth.

**Figure 3 microorganisms-12-01203-f003:**
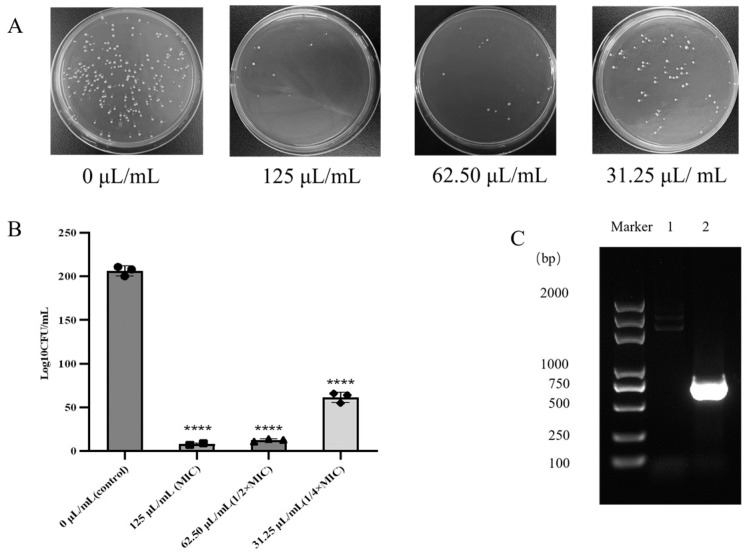
Plasmid binding transfer. (**A**) Growth of CRKP co-cultured with 125 μL/mL (MIC), 62.50 μL/mL (1/2× MIC), and 31.25 μL/mL (1/4× MIC) FB1-1 CFS for 24 h. (**B**) CRKP was co-cultivated with FB1-1 CFS at concentrations of 125 μL/mL, 62.50 μL/mL, and 31.25 μL/mL for 24 h; the viable bacterial count was determined. (**C**) Agarose gel electrophoresis after amplification of plasmid transconjugant *bla_KPC* DNA. Lane 1 is the vector *E. coli* EC600, and Lane 2 is the plasmid transconjugant. ****, *p* < 0.0001 compared to control group.

**Figure 4 microorganisms-12-01203-f004:**
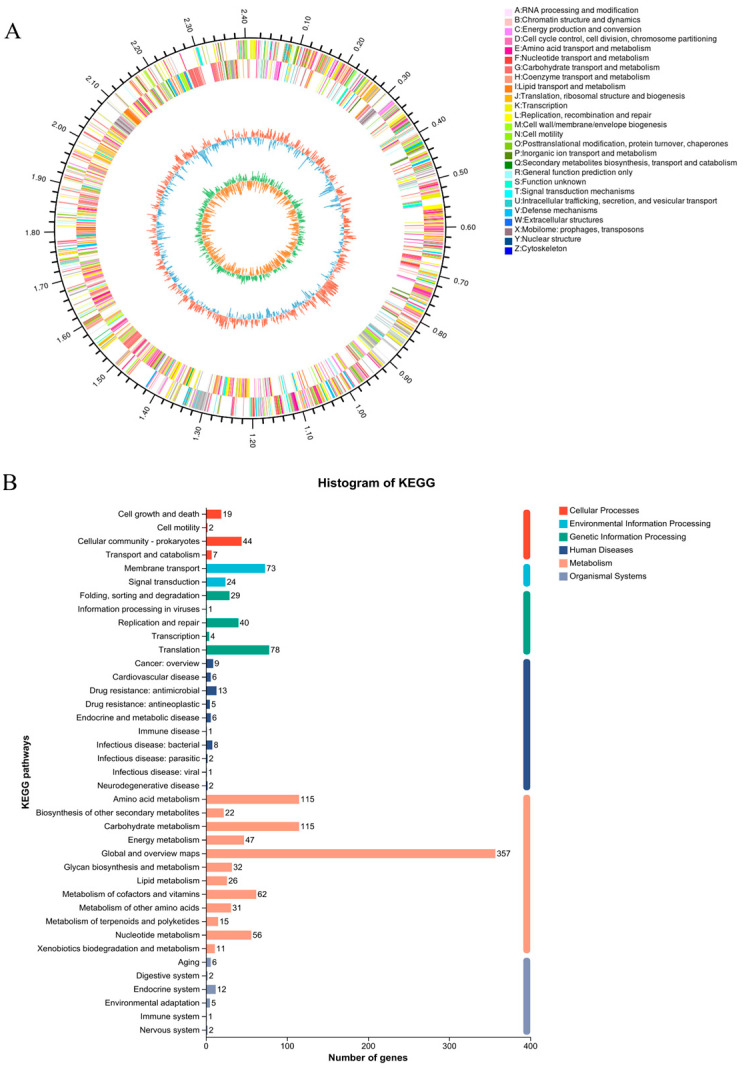
Gene sequence and information annotation. (**A**) Circos genome circle diagram: the outermost circle of the diagram denotes genome size. The second and third circles represent CDS on positive and negative chains, with colors signifying the functional classification of distinct CDS cogs. The fourth circle denotes rRNA and tRNA, while the fifth circle displays GC content. (**B**) KEGG annotation: the ordinate indicates the level-2 classification of KEGG pathways, and the abscissa indicates the gene count for each classification. Different column colors represent level-1 hierarchical classifications of KEGG pathways.

**Figure 5 microorganisms-12-01203-f005:**
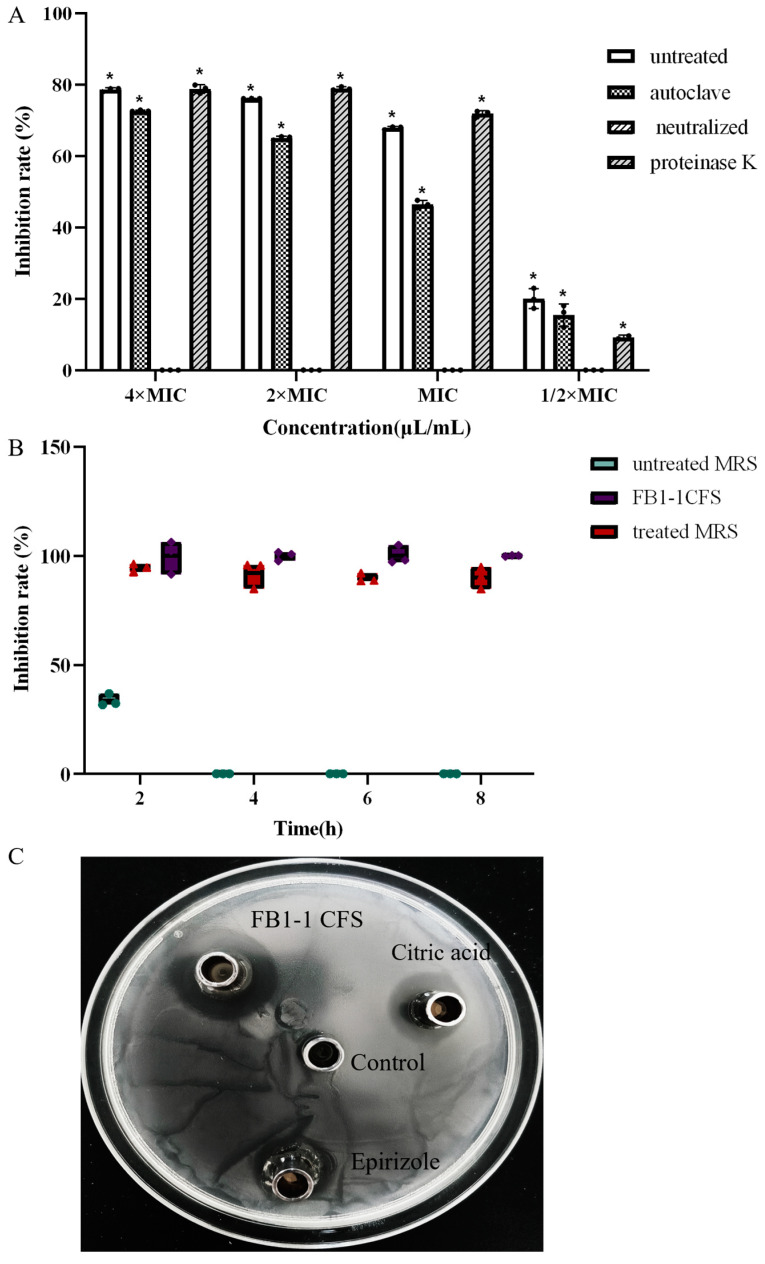
Analysis of antibacterial substances. (**A**) The inhibition percentage of 10^6^ CFU/mL CRKP co-cultured with different concentrations (4× MIC, 2× MIC, MIC, and 1/2× MIC) of FB1-1 CFS were treated with heating (autoclave), proteinase K, and NaOH neutralization. *, *p* < 0.05 compared to neutralized CFS. (**B**) CRKP was co-cultured with both treated MRS medium, which had been pH-adjusted to match that of FB1-1 CFS, and with untreated MRS medium as well as FB1-1 CFS itself. Samples were collected at 2, 4, 6, and 8 h post cultivation to determine the antibacterial activity rates. (**C**) The control group was cultured with MRS medium, while the experimental groups were treated with FB1-1 CFS, citric acid, and epirizole, respectively.

**Table 1 microorganisms-12-01203-t001:** qRT-PCR primer sequences used in this study.

Primer Name	Primer Sequence (5′-3′)
16S rRNA_F	AGAGTTTGATCCTGGCTCAG
16S rRNA_R	TACGACTTAACCCCAATCGC
*bla_KPC*F	CGCTGTGCTTGTCATCCTTG
*bla_KPC*R	GGCACGGCAAATGACTATGC
*uge*_F	GGAAGGCTGCCGTCATACCA
*uge*_R	GGATTACATCCTGCACCCGAAC
*fim_H* _F	AGGATCGTTAATCGCGGTGCTG
*fim_H* _R	TGTGGTCCGGGCATAGGTGG

**Table 2 microorganisms-12-01203-t002:** Potential antibacterial compounds.

Compound	Retention Time	% of Total
Citric acid	1.097	16.56
Epirizole	1.146	3.18
DL-arginine	0.795	2.65
Betaine	0.828	1.50
7-Diethylaminocoumarin-3-carboxylic acid	2.905	0.89
Gamma-aminobutyric acid	0.798	0.07
2-Hydroxyquinoline	1.381	0.05
Kojic acid	3.839	0.05
Anthranilic acid	3.141	0.01

## Data Availability

The raw data supporting the conclusions of this article will be made available by the authors on request.

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
