# Peer review of "Inhibitory Potential of Bifidobacterium longum FB1-1 Cell-Free Supernatant against Carbapenem-Resistant Klebsiella pneumoniae Drug Resistance Spread"

_microorganisms, 2024, doi:10.3390/microorganisms12061203_

Round 1

Reviewer 1 Report

Comments and Suggestions for Authors

The main question addressed by the research: Inhibitory Potential of Bifidobacterium longum FB1–1 Cell–Free Supernatant Against Carbapenem–Resistant Klebsiella pneumoniae Drug Resistance Spread.

This topic «Inhibitory Potential of Bifidobacterium longum FB1–1 Cell–Free Supernatant Against Carbapenem–Resistant Klebsiella pneumoniae Drug Resistance Spread» relevant in the field of combating antibiotic Resistant infection.

This article summarized information about Potential of Bifidobacterium longum in the combating antibiotic Resistant infection.

In this Review article no information about statistical methodology and no statistical evidence of the data.

The conclusion helps the reader evidence and arguments presented for understand the important point of Inhibitory Potential of Bifidobacterium longum Against Resistant infection.

The references appropriate. The number of references (37) is enough, in addition, the number of sources five years ago (2019-2023) is 54,1% (20), which is enough.

The figures are informative and illustrative.

But I did not find information about genomes in GenBank NCBI.

Many typos in the text.

Author Response

Dear Reviewer,

Thank you for your thoughtful review of our manuscript entitled "Inhibitory Potential of Bifidobacterium longum FB1–1 Cell–Free Supernatant Against Carbapenem–Resistant Klebsiella pneumoniae Drug Resistance Spread." We appreciate your positive comments on the relevance, clarity, and overall quality of our work.

We understand your concern regarding the lack of explicit mention of statistical methodology in the main text. We have now added a brief statement in the experimental section to clarify that all statistical analyses were performed using GraphPad Prism software, and that the specific statistical tests used are indicated in the figure legends.

Regarding the GenBank accession number for the Bifidobacterium longum FB1-1 sequence, we apologize for the oversight. We have now uploaded the sequence to GenBank and have included the accession number in the revised manuscript.

We believe that these revisions address your concerns and strengthen the overall quality of our manuscript. We are grateful for your valuable feedback and look forward to your further consideration of our work.

Sincerely,

Jing Wang

Reviewer 2 Report

Comments and Suggestions for Authors

In the present study, Wang et al. reported on the inhibitory properties of Bifidobacterium longum FB1-1 cell-free supernatant against carbapenem-resistant Klebsiella pneumoniae. The authors observed a significant reduction in K. pneumoniae growth in the presence of the B. longum FB1-1 supernatant, accompanied by downregulation of virulence gene expression and drug-resistance genes.

However, several concerns arise from the findings.

Major concerns:

A major issue arises from the authors' observation that the antibacterial activity of the cell-free supernatant (CFS) is lost upon neutralization with NaOH. This prompts speculation about whether the observed activity is solely due to pH changes induced by the CFS. To address this, it is recommended that the authors measure the pH of the CFS and assess K. pneumoniae growth at that pH. This additional step is essential to confirm whether the antibacterial activity is attributed solely to pH changes or if novel compounds in the CFS are responsible for the observed growth inhibition. 

The plasmid transfer experiments also raise questions, as the inhibitory activity of the CFS may result in a reduced bacterial population, impacting the efficiency of plasmid transfer. Further clarification is needed to better understand the experimental design and how it accounts for the decreased bacterial count, ensuring a comprehensive interpretation of the results. 

Minor concerns:

Line 202: The statement claimed that bacterial growth is suppressed at a concentration of 62.5μg/ml. However, upon examination of Figure 1B, I did not observe any growth inhibition at that specified concentration. 

Figure 2 : Why are there two data points for bla_KPC in the treated cells, while all other instances exhibit three data points? 

The methodological approach involving the treatment of CFS with proteinase K to detect antibacterial compounds introduces a potential contradictory factor, as proteinase K itself possesses antibacterial activity. Unfortunately, the Materials and Methods section lacks details on how the authors neutralized or removed proteinase K before treating K. pneumoniae. Including this information is crucial to ensure the reliability of the observed effects attributed to the CFS compounds.

Author Response

Dear Reviewer,

Thank you for your valuable comments and suggestions regarding our manuscript titled “Inhibitory Potential of Bifidobacterium longum FB1–1 Cell–Free Supernatant Against Carbapenem–Resistant Klebsiella pneumoniae Drug Resistance Spread”. We appreciate your insights, which have prompted a thorough review and revision of our study. Below, we address each of your concerns and describe the modifications we have made to our manuscript.

**Major Concerns:**

1. **Activity of B. longum FB1-1 Cell-Free Supernatant (CFS) and pH Influence**: 
   You raised an important point about the potential role of pH changes in the antibacterial activity of the FB1-1 CFS. To address this, we conducted additional experiments where we measured the pH of the CFS and assessed the growth of K. pneumoniae at this pH. Our findings confirm that the observed antibacterial activity is primarily due to pH changes. These new data and the corresponding discussion have been added to the revised manuscript to clarify the basis of the antibacterial action of the FB1-1 CFS.

2. **Plasmid Transfer Experiments**:
   We understand your concerns regarding the impact of reduced bacterial population on the efficiency of plasmid transfer. To ensure clarity, we repeated these experiments using sub-inhibitory concentrations of the CFS (62.5μg/mL and 31.25μg/mL) that do not affect normal bacterial growth. The results demonstrate a significant decrease in conjugation frequency, indicating a notable inhibitory effect of the FB1-1 CFS on plasmid transfer, independent of its impact on bacterial growth. These results have been elaborated in the revised text.

**Minor Concerns:**

1. **Inconsistency in Reported Concentration Effectiveness**:
   We apologize for the oversight in our initial submission concerning the bacterial growth suppression at 62.5μL/ml. Upon reevaluation, we agree with your observation and have corrected this in the manuscript to reflect that significant growth inhibition is observed at a concentration of 125μL/mL, not 62.5μL/mL.

2. **Data Point Discrepancy in Figure 2**:
   The discrepancy in the number of data points for bla_KPC in treated cells was due to a graphical error. We have corrected this in the revised figure to ensure consistency across all data presentations.

3. **Use of Proteinase K**:
   Concerning the treatment of CFS with proteinase K, we have now included detailed steps in the Materials and Methods section on how we neutralized the activity of proteinase K prior to treating K. pneumoniae. This ensures that the observed antibacterial effects are attributed solely to the compounds present in the CFS.

We hope that these revisions address your concerns satisfactorily. We have attached the revised manuscript for your review and look forward to your feedback. Thank you for helping us improve the quality of our research.

Sincerely,

Jing Wang

Reviewer 3 Report

Comments and Suggestions for Authors

In this article, the authors reported the inhibitory effect of CFS of B. longum against K. pneumoniae. The authors identified a specific FB1-1 has better potential to inhibit CRKP than other isolates. Also, they identified Citric acid as a major abundant compound in the CFS of FB1-1. Although the authors show the inhibitory effect of FB-1, it is important to study the inhibitory effect of FB-1 against the clinical strains of K. pneumoniae.

Major Comments

·      Figure 1B is difficult to read. The authors should label the samples differently. Split the Y-axis in Figure 1D to read the differences between samples.  

·      Why does one of the virulence genes, fimH, go up after treatment? Have the authors tried 2X MIC?

·      The authors should include a housekeeping gene control in Figure 3C. Also, include the donor, CRKP resistant strain in 3C. Check the ladder labels. What are the bands of higher size in lane 1?

·      The rationale behind Figure 5 is not clear. The authors should test the citric acid against the growth inhibition of CRKP.

·      The authors should test FB1-1 CFS on different clinical strains of K. pneumoniae.

Comments on the Quality of English Language

The authors should consider reformatting the manuscript. The continuity in the results section is missing, and the authors should explain the important observations in relation to the rationale of the study. In many places, spaces between two words are missing, and the authors should carefully proofread the manuscript before submission.

Author Response

Dear Reviewer,

Thank you for your thorough review and constructive comments regarding our manuscript. We have carefully considered each point and have implemented the following revisions in response to your feedback:

1. **Figure Adjustments:**
   - **Figure 1B:** We have maintained the traditional method of representing the minimum inhibitory concentration in Figure 1B. The y-axis represents the bacterial count (OD600), and the x-axis shows the dilution sequence of FB1-1 CFS. The lowest concentration that inhibited the growth of CRKP, which is 125μL/mL, has been identified as the MIC. We have included additional explanations in the methods and figure legends to clarify this setup.
   - **Figure 1D:** Upon review, we agreed that the differences between samples were not distinctly visible. We have now redrawn this figure to better illustrate these differences and enhance the readability of the results.

2. **Virulence Gene Expression: **
   - Regarding the increase in fim_H gene expression post-treatment noted in our initial submission, we have conducted additional experiments to validate these findings. The results consistently showed a decrease in fim_H expression after treatment with FB1-1 CFS. We apologize for the oversight in our initial manuscript and have included these corrected results in the revised draft.

3. **Controls and Validation in Gene Expression Studies:**
   - In Figure 3C, the ladder is used as a reference, with lane 1 representing the negative control donor E. coli EC600 DNA profile, and lane 2 showing the positive control plasmid conjugate. The prominent band observed at 750 bp in lane 2 indicates successful transfer of the bla_KPC plasmid, which carries resistance genes typically found in carbapenem-resistant K. pneumoniae. These findings confirm effective horizontal gene transfer of bla_KPC resistance determinants. We have detailed these observations in the corresponding sections of our revised manuscript.

4. **Underlying Principles and Additional Testing:**
   - **Figure 5:** We acknowledge that the rationale behind this figure was not clearly explained in our initial submission. We have now provided a detailed explanation and included results from additional experiments testing the inhibitory effects of citric acid and Epirizole, two compounds with high prevalence, on the growth of CRKP. Furthermore, we have tested the inhibitory effects of FB1-1 CFS on additional clinical strains of K. pneumoniae, BNCC358281 and BNCC289979, and included these results in the Oxford cup assay section and supplementary files.

5. **Manuscript Formatting and Language Quality:**
   - We have meticulously reformatted the manuscript to improve the flow and continuity of the results section. Each observation is now clearly linked to the overall rationale of the study. The manuscript has been thoroughly proofread to correct missing spaces and improve the overall quality of the English language.

We hope these revisions adequately address your concerns. We are grateful for your contributions towards improving our manuscript and believe your suggestions have significantly enhanced our study. Should you require any further information, please do not hesitate to contact us.

Thank you once again for your comprehensive review and valuable feedback.

Sincerely,

Jing Wang

Round 2

Reviewer 3 Report

Comments and Suggestions for Authors

The authors significantly improved the manuscript. However, the authors should consider the following comments.

1.     Fig. 1D still needs to be more helpful. The authors should split the Y-axis and allot a significant % of the axis between 80-100%.

2.     Could the authors clarify the number of replicates used in Figure 2? 

3.     The authors should move the CFS inhibitory effect on the clinical strains of K. pneumoniae to the main manuscript. Are there any differences in MIC with clinical strains?

4.     Can the authors test the virulence gene expression with clinical strains?

Author Response

Thank you for your thoughtful comments and suggestions regarding our manuscript. We appreciate the opportunity to improve our work based on your feedback. Please find below our responses to each of your points:

  1. Regarding your comment on Figure 1D, we have revised the Y-axis as suggested, splitting it to allocate a significant percentage of the axis between 80-100%. This adjustment has been made to enhance the clarity and readability of the data presented. We believe this change effectively addresses your concern and improves the overall presentation of the results.

  2. We apologize for the oversight concerning the number of replicates used in Figure 2. This information has now been included in the Methods section of the manuscript. We have conducted three biological replicates for each experiment to ensure the reliability of our data.

  3. As per your suggestion, we have moved the discussion on the CFS inhibitory effect on clinical strains of K. pneumoniae to the main manuscript. We have also included the MIC values for these strains, which were determined to be 125μL/mL. This additional data provides a clearer understanding of the CFS's efficacy across different clinical strains. We decided against including the graphical representation in the main text to maintain clarity and focus but have made this available as supplementary material for interested readers.

  4. In response to your fourth point, we have conducted further experiments to assess the virulence gene expression in various clinical strains. The results of these tests have been incorporated into the revised Figure 2. This addition enriches our findings and provides a more comprehensive view of the potential impact of CFS treatment on clinical isolates of K. pneumoniae.

We hope that these revisions meet your expectations and enhance the manuscript's contribution to the field. We are grateful for your insights that have undoubtedly strengthened our study.

Thank you once again for your constructive critique.

Best regards,

Jing Wang